# Pre-Harvest Supplemental Blue Light Enhanced Antioxidant Activity of Flower Stalk in Chinese Kale during Storage

**DOI:** 10.3390/plants10061177

**Published:** 2021-06-09

**Authors:** Haozhao Jiang, Xia Li, Jialing Tian, Houcheng Liu

**Affiliations:** College of Horticulture, South China Agricultural University, Guangzhou 510642, China; jhzh111@stu.scau.edu.cn (H.J.); xial@stu.scau.edu.cn (X.L.); jialingt@outlook.com (J.T.)

**Keywords:** *Brassica alboglabra*, pre-harvest supplemental blue light, antioxidant activity, nutritional quality, storage

## Abstract

For 10 days before harvest, supplemental 50 μmol m^−2^ s^−1^ blue light (430 nm) was applied in greenhouse conditions in order to evaluate the influences of pre-harvest supplemental blue light on both antioxidants and nutrition of the flower stalk of Chinese kale during storage. The weight loss of the flower stalk of Chinese kale treated with supplemental blue light was generally lower than control during storage. Higher antioxidant activity was maintained during storage by supplemental blue light. Meanwhile, supplemental blue light derived higher contents of vitamin C, soluble protein, free amino acids, and chlorophyll at harvest. The samples exposed to supplemental blue light possessed both higher nutrition and antioxidant values. Thus, pre-harvest supplemental blue light treatment might be a promising strategy to enhance the antioxidant activity and nutritional values and extend the shelf-life of the flower stalk of Chinese kale.

## 1. Introduction

Chinese kale (*Brassica alboglabra* Bailey) is considered an indigenous vegetable in Guangzhou, China. The flower stalk is the main edible part, which is crisp and rich in health-promoting phytochemicals (e.g., vitamin C, glucosinolates, and phenolics) [1]. The greater the consumption of these bioactive compounds, the less the risk of degenerative diseases [2,3]. Leaf senescence, which might lead to nutritional and commercial loss, has been the predominant problem for most leaf vegetables during storage. The same is true of post-harvest Chinese kale [4,5,6]. Those deteriorations might be ascribed to multiple underlying processes that usually happen simultaneously [7], and these processes could be induced by light [8].

Light has been proved to maintain and enhance the nutritional quality of foods in their post-harvest stage [9]. Light exposure could slow down the speed of browning in fresh-cut romaine lettuces [10], fresh-cut celery [11], and post-harvest broccoli [12]. With fluorescent light treatment, post-harvest Chinese kale lost weight and vitamin C content but reached higher starch, fructose, and glucose contents [5]. Post-harvest pulsed light treatments (20 kJ m^−2^ and 60 kJ m^−2^) demonstrated significant effects on soluble solids, total phenols, and antioxidant capacity in persimmons [13]. A higher content of vitamin K_1_ was revealed in spinach top-canopy leaves under continuous post-harvest light treatment than those in darkness [14]. Except for glutamic and aspartic acids, the remaining free amino acids of post-harvest tomatoes treated with blue light were higher than those in darkness or red light treatment [15]. In post-harvest brussels sprouts, white-blue light treatment retarded senescence and remarkedly improved total flavonoids content compared with dark treatment [16]. A similar result was found in broccoli during its post-harvest life [4]. In strawberries, post-harvest blue light treatment enhanced the anthocyanin content by activating related enzymes [17]. Red light massively retarded the etiolation, ethylene synthesis, and ascorbic acid reduction of broccoli after harvest compared with blue light [18]. Among five wavelength LEDs (blue/green/yellow/red/white) treatments, the contents of chlorophyll and ascorbic acid in post-harvest broccoli was highly increased after green light treatment [19]. Post-harvest UV treatment could strengthen antioxidant activities and improve the phenolic content of vegetables and fruits (e.g., table grapes [20], broccoli florets [21], and blueberries) [22]. Light quality as an energy source prospectively drives the responses of vegetables to post-harvest light treatments.

A few studies have revealed how light treatments affected the post-harvest quality of plant products during either the vegetative period or pre-harvest period. In greenhouses, supplemental white fluorescent tubes at the growth period could extend the shelf-life and enhance the post-harvest qualities of lettuce [23]. UV–B radiation (280−315 nm) in the vegetative period strongly inhibited weight loss and observably increased the total phenolic and total flavonoid contents of broccoli during storage [24]. Exposure to UV–B irradiation during the growth stage could effectively improve the total phenolic content and maintain the freshness of mung bean sprouts [25]. Pre-harvest UV–B application remarkedly promoted the polyphenols content and improved the antioxidant activity of stored basil leaves without detrimental effects on their visual quality [26]. In Chinese kale sprouts, obvious increases of the content of vitamin C and total phenolic and antioxidant activity exposed by pre-harvest red light during storage were reported [27]. However, the effect of pre-harvest light quality on health-promoting nutrition in vegetables during storage remains to be further elucidated.

In this study, we investigated how pre-harvest supplemental blue light affected weight loss, chlorophyll content, health-promoting nutrition, and antioxidant activity in the flower stalk of Chinese kale during storage.

## 2. Results

The morphology in the flower stalk of Chinese kale during storage showed no significant differences between the two treatments. However, those flower stalks exposed to supplemental blue light indicated slower browning than the control stalks during storage (Figure 1).

### 2.1. Weight Loss

The weight loss in the flower stalk of the Chinese kale markedly increased during storage (Figure 2). Although no significant difference was observed between the two treatments, the weight loss of flower stalks treated with blue light were generally lower than control during storage. At 6 DAH, the flower stalk’s weight loss was 4.7% and 4.3% in the control and blue light treatment, respectively. Hence, pre-harvest supplemental blue light treatment contributed to an extended shelf-life of the flower stalk of Chinese kale.

### 2.2. Pigment Content

Regardless of pre-harvest treatment, the contents of chlorophyll a, chlorophyll b, and total chlorophyll observably decreased during storage (Figure 3). At 6 DAH, the total chlorophyll content was about 50% less than at harvest. At 0 DAH, the flower stalk exposed to blue light indicated higher contents of chlorophyll a, chlorophyll b, and total chlorophyll than the control, with increases of 11.5%, 18.2% and 13.1%, respectively. No significant differences in these pigment contents were observed between the two treatments during storage. The carotenoid content in the flower stalk showed few changes during storage. Although a lower carotenoid content in flower stalk supplemented with blue light than control at harvest was found, there were no differences in carotenoid content between treatments during storage.

Hence, supplemental blue light treatment contributed to the maintenance of pigment contents in the flower stalk of Chinese kale during storage.

### 2.3. Soluble Protein and Free Amino Acid Content

The soluble protein content in the flower stalk of Chinese kale decreased during storage (Figure 4). At harvest, a higher proportion of soluble protein content (33.6%) in the flower stalk was observed in the supplemental blue light treatment than in control. There was a greater decrement of soluble protein content during storage in blue light treatment compared to control, which reduced 18.3% at 4 DAH.

The free amino acid content in the flower stalk massively increased in both treatments during storage (Figure 4), with increases of 180.6% and 145.6% in supplemental blue light and control, respectively, at 6 DAH compared to 0 DAH. The free amino acid content in the flower stalk of the supplemental blue light treatment was 21.1% and 20.4% lower than control at 2 and 4 DAH, respectively. However, supplemental blue light stimulated a higher production of free amino acid content, whereas control was about 33.4% higher at 6 DAH.

The exposure to supplemental blue light highly indicated the contents of soluble protein and free amino acid in the flower stalk of Chinese kale during storage.

### 2.4. Vitamin C Content

The vitamin C content in the flower stalk of Chinese kale presented sharply decreased, about 20.1% in control and 53.4% in supplemental blue light treatment at 2 DAH, respectively (Figure 5). The vitamin C content in the flower stalk treated with supplemental blue light was 9.1% lower than the control at 4 DAH, while were 16.3% and 16.1% higher than the control at 0 and 6 DAH, respectively. It was feasible that supplemental blue light treatment could effectively maintain Vitamin C content in the flower stalk of Chinese kale during storage.

### 2.5. Total Phenolic and Flavonoids Contents

Pre-harvest blue light enhanced the total phenolic and flavonoids contents in the flower stalk of Chinese kale during storage (Figure 6). The total phenolic content in the control flower stalk decreased 26.9% at 2 DAH, and remained steady during storage. Those treated with supplemental blue light revealed no marked decrease during storage. Even though the total phenolic content in the flower stalk treated with supplemental blue light was lower (10.6%) at harvest, this was 19.7%, 13.3%, and 10.1% higher than control at 2, 4, and 6 DAH, respectively. Hence, pre-harvest supplemental blue light contributed to the maintenance of the total phenolic content.

The flavonoids content in the flower stalk of Chinese kale decreased 109.6% in control at 2 DAH, while those in supplemental blue light treatment barely changed. Those treated with supplemental blue light were 18.3% and 109.6% higher than the control at 2 and 4 DAH, respectively. The flavonoids content exhibited a 51.4% and 40.13% decline in both treatments at 6 DAH compared to 0 DAH. Overall, the reduction of the flavonoids content treated with supplemental blue light was lower than control.

### 2.6. Antioxidant Capacity Assay

Both the FRAP value and DPPH activity were involved in evaluating the total antioxidant capacity in the flower stalk of Chinese kale (Figure 7). The FRAP value (16.7%, 30.7% and 8.9%) and DPPH activity (0.6%, 0.5% and 1.5%) were higher in supplemental blue light treatment compared to control from 2 DAH to 6 DAH. Hence, pre-harvest supplemental blue light maintained the antioxidant capacity in the flower stalk of Chinese kale during storage.

### 2.7. Multivariate Principal Component Analysis

To compare the correlation of all quality traits in the flower stalk of Chinese kale in supplemental blue light treatment and control during storage, the principal component analysis (PCA) was performed (Figure 8 and Table 1). The first five principal components (PC1–PC5) were associated with eigen values > 1, in order to account for 94% and 100% of the cumulative variance (Table 1).

The first two factors (PC1 vs. PC2) of the PCA were presented and revealed 76.6% of the total variance in the flower stalk of Chinese kale in supplemental blue-light treatment and control during storage (Figure 8). PC1 was positively correlated to FRAP, DPPH, flavonoids (fla), and other antioxidant activities. Vitamin C (VC) and weight loss (WL) were negatively correlated to PC2. The outcomes indicated the relationship among nutrition, pigment and antioxidant components by identifying the angle between two vectors (0° < positively correlated < 90°; uncorrelated, 90°; 90° < negatively correlated < 180°). Strong positive correlations were found between chlorophylls (a, b, t), carotenoids (c), total phenolic (phe), flavonoids, soluble protein (pro), DPPH activity, and FRAP value in the flower stalk of Chinese kale during storage, as their angles were less than 90°. Both blue light treatment and control showed in different quadrants, which means that these two treatments were separated clearly.

### 2.8. Heatmap Assay

A heatmap synthesizing the response of the measured parameters provided an integrated view of the effect of pre-harvest supplemental blue light treatment on the maintenance of the quality of the flower stalk in Chinese kale (Figure 9).

The cluster exhibited different nutritional quality and antioxidant activity in the flower stalk of Chinese kale in both supplemental blue light treatment and control during storage. The flower stalk of Chinese kale in control revealed more weight loss during storage, and higher contents of free amino acids and Vitamin C at 2 and 4 DAH. However, the flower stalk exposed to supplemental blue light presented lower total phenolic content at 0 DAH and flavonoids content at 6 DAH, revealed higher total phenolic content at 2, 4, and 6 DAH, had higher contents of chlorophyll, carotenoids, Vitamin C, and flavonoids, as well as higher values of DPPH and FRAP during storage compared to control. These results indicate that pre-harvest supplemental blue light contributed to the maintenance of quality in the flower stalk of Chinese kale during storage. 

## 3. Discussion

Stomatal transpiration in detached leaves and plants is strongly related to weight loss, which could be regulated by light during either pre-harvest or post-harvest (especially blue light) [28]. During storage at 1 °C, the fresh weight loss of Chinese kale was higher in light treatment using fluorescent tubes than in dark, which was positively correlated with stomata opening [5]. The weight loss of broccoli during the storage period increased by the continuous white-blue light radiation due to a higher transpiration rate [4]. The weight loss of broccoli heads showed an increase in conjunction with continuous low intensity white light treatment during post-harvest storage [29]. On the contrary, the weight loss of broccoli was strongly inhibited by the pre-harvest UV–B radiation compared with the control [24]. In this study, the flower stalk of Chinese kale lost 4.7% and 4.3% in control and supplemental blue light treatment at 6 DAH, respectively, which meant a lower weight loss in the flower stalk radiated by pre-harvest supplemental blue light (Figure 2). However, no remarkable difference between treatments was found in this study, which might be due to the fact that there were less of an effect on stomata opening under pre-harvest supplemental blue light than post-harvest light treatment.

Chlorophyll content is one of the typical physiological indexes related to senescence [30]. Chlorophyll degradation is strongly linked to the lipid oxidation of cell membranes and the antioxidant enzyme activity of plants. Light exposure could trigger specific physiological processes in vegetables which might result in pigment changes [31]. The chlorophyll content of greenhouse pak choi increased with a higher proportion of supplemental blue light at harvest [32]. Blue light (50 μmol m^−2^ s^−1^) slightly inhibited chlorophyll degradation in pak choi during storage [33]. However, at the same light intensity single blue light reduced the carotenoids content of Chrysanthemums compared with combination of red and blue light [34]. Blue light (430 nm and 465 nm) markedly improved the content of both chlorophylls and carotenoids in Chinese kale and pak choi baby-leaves, while 430 nm significantly increased the chlorophyll content and 465 nm revealed the highest carotenoids content [35]. In this study, the contents of chlorophyll a, chlorophyll b, and total chlorophyll in the flower stalk of Chinese kale decreased both in supplemental blue light treatment and control during storage (Figure 3). The chlorophyll content of the flower stalk of Chinese kale was also highly induced by pre-harvest supplemental blue light treatment during storage. These results indicate that pre-harvest supplemental blue light contributed to maintain pigment contents of the flower stalk of Chinese kale during storage.

Soluble protein is considered to be a sensitive osmotic regulator and nutrition to regulate metabolism, which is greatly responsive to light. Chloroplast proteins degradation is one of the characteristic symptoms which would be extensively activated during senescence. A reduction of soluble protein content was found in both post-harvest light treated and control broccoli during storage [12]. In this study, pre-harvest supplemental blue light treatment showed a higher soluble protein content than control in the flower stalk of Chinese kale during storage (Figure 4). Soluble protein degradation was accompanied by chlorophyll degradation in Chinese kale after harvest, which was also the case in tobacco leaves [36]. It is feasible that pre-harvest supplemental blue light increased the soluble protein content in the flower stalk of Chinese kale and delayed senescence.

The taste of vegetables is partially influenced by their free amino acids, which come from proteolysis [37]. Amino acid metabolism plays an important role in stress responses and some secondary metabolites are further derived from various amino acids [38]. The cultivation with blue LEDs showed a higher concentration of free amino acids in *Spirulina* sp. cultures [39]. In this study, pre-harvest supplemental blue light markedly induced the flower stalk of Chinese kale to enhance free amino acids content and decrease soluble protein content during storage (Figure 4). The content of free amino acids in the flower stalk of Chinese kale in control was higher at early storage period, while those in supplemental blue light treatment ended up with higher free amino acid content at 6 DAH, which might be highly correlative to different proteolysis and other secondary metabolisms during storage.

Vitamin C, also known as ascorbic acid, is a necessary micronutrient for the human body (especially for the functioning of bodily systems). The de novo biosynthesis, degradation, and recycling of vitamin C collectively keep balance in plants. In three citrus varieties, blue LED light treatment highly increased the vitamin C content as well as transcription expression of genes related to vitamin C biosynthesis [40]. A significant reduction of vitamin C content in the flower stalk of Chinese kale was found in two treatments at 2 DAH (Figure 5). The vitamin C content in harvested broccoli reduced in darkness during storage because of lower stability of vitamin C, owing to a high ROS content [41]. In red and green leaf pak choi, vitamin C content showed a marked increase by supplemental blue light [32]. Pre-harvest supplemental blue light treatment increased vitamin C content in the flower stalk of Chinese kale at 0 and 6 DAH compared with the control (Figure 5). Thus, pre-harvest supplemental blue light might increase the expression of biosynthesis, enhance the regeneration of genes related to vitamin C, and effectively keep the vitamin C content in the flower stalk of Chinese kale during storage.

The antioxidant properties of each food matrix come from the combined and concerted action of biologically active compounds (i.e., polyphenols, carotenoids, lignans, glucosinolates, etc.) [42]. Phenolic compounds, including flavonoids, are important secondary metabolites in plants. The phenolic content in plants relies on the balance between their synthesis via phenylalanine ammonia pathways, in which PAL is the key enzyme and oxidation occurs by PPO and POD [43]. In this study, higher contents of total phenolic and flavonoids content were revealed in pre-harvest supplemental blue light treated Chinese kale during storage compared with the control because of lower degradation (Figure 6). In post-harvest broccoli, the total phenolic and total flavonoids contents decreased due to degradation reactions. However, higher contents were revealed in broccoli treated by pre-harvest UV during storage [41]. The transcription expression of phenylpropanoid biosynthesis genes and phenolic compounds were both higher in tartary buckwheat sprouts grown under blue LEDs than in those grown under white or red LEDs [44]. In previous studies, a higher level of total phenolic content was also found in strawberry fruit treated with post-harvest blue light after two days of storage [17]. The highest level of total phenolic compounds was obtained in blue light-radiated canola sprouts [45]. Through photoreceptors and genes, plants regulate the biosynthesis of different secondary metabolites in respond to light. For instance, the combined blue and red light promoted the growth of *S**alvia miltiorrhiza* and enhanced the accumulation of phenolic acids by upregulating the transcription of SmPAL1 and Sm4CL1 in this herb [46]. Light signals are recognized by distinct photoreceptors; cryptochromes and phototropins recognize blue wavelength. Cryptochrome action could be triggered by blue light recognized by flavin adenine dinucleotide and methenyltetrahydrofolate chromophores [47]. Blue light upregulated the expression of flavonoid biosynthesis-related genes (PAL and F3′H) and induced flavonoid biosynthesis in *Cyclocarya paliurus*. Furthermore, blue light could induce cryptochrome action and trigger anthocyanin biosynthesis via accelerated PAL activity [48]. The enzyme behind flavonoid biosynthesis is chalcone synthase (CHS), and it is shown that blue light inductions in CHS expression are mediated by cryptochrome (cry1) [49]. Even though pre-harvest supplemental blue light treatment reduced the total phenolic and flavonoids content in the flower stalk of Chinese kale at 0 DAH, the total phenolic content in blue light treatment exhibited higher than control at 2, 4, and 6 DAH, respectively. Supplemental blue light could trigger the cryptochrome, which could activate the expression of corresponding genes (i.e., Cry and F3′H) and enzymes (i.e., PAL and CHS), and lead to the increased biosynthesis of antioxidants during post-harvest life. Hence, pre-harvest supplemental blue light could maintain the total phenolic and flavonoids content in the flower stalk of Chinese kale during storage.

Vitamin C, flavonoids, and phenolics are major antioxidants in vegetables, which are highly related to antioxidant activity [50]. Antioxidant activity was bound up with FRAP and DPPH. FRAP assays and DPPH radical scavenging activity were employed to evaluate the antioxidant activity of Chinese kale in this study. These two measurements could be used to assay the ability to limit, reduce, or remove oxidation. Pre-harvest supplemental blue light treatment enhanced the antioxidant activity of the post-harvest flower stalk of Chinese kale (Figure 7). A similar result was found in common buckwheat sprouts; blue light increased both the total phenolic content and the total flavonoids content, as well as antioxidant activities [51]. It has been showed that higher DPPH radical scavenging activity was observed in post-harvest blue light-treated strawberry fruits than those in control fruits after 4 d of storage [17]. In this study, the FRAP value was strongly correlated to total phenolic content, total flavonoids content, and DPPH radical scavenging activity (Figure 8). Obviously, higher total phenolic and flavonoids content, as well as the FRAP value and DPPH activity, were maintained in the flower stalk of Chinese kale during the post-harvest storage by pre-harvest supplemental blue light.

## 4. Materials and Methods

### 4.1. Plant Material and Cultivation Conditions

The experiment was carried out in the greenhouse of the South China Agricultural University with natural light conditions. The seeds were sown in perlite with 1/4 strength Hoagland nutrient solution. After 15 d, the seedlings with three expended true leaves were transplanted into 10 L planting bag filled with 1:1 perlite:coconut coir (V:V). During the whole growth period, nutrient solution was applied once a day in the morning using 1/2 strength Hoagland nutrient solution.

### 4.2. Supplemental Blue Light Treatment

The planting bags were separated into two groups 30 days after transplantation. One group was supplemented with blue LED light (430 ± 10 nm, 50 μmol m^−2^ s^−1^, 7:00−19:00; Chenghui Equipment Co., Ltd., Guangzhou, China), while the other group was used as control.

### 4.3. Storage of Chinese kale

After 10 days of blue light treatment, the Chinese kale was harvested. The flower stalks were packed into commercial polypropylene packages, with 15 bags per treatment. After measuring the weight of each bag, they were placed on plastic tray and kept in a dark storage room with an average temperature of 15 ℃ and a relative humidity of 85%. Every three samples were assayed at 0, 2, 4, and 6 d after harvest (DAH). The samples were frozen immediately in liquid nitrogen and lyophilized at −80 ℃ freezer for quality analyses.

### 4.4. Weight Loss

At harvest, three individually numbered bags from each group as three replicates were weighted, and re-weighted at 2, 4, and 6 d after harvest. Weight loss was presented as percent lost from the initial weight, according to the following equation: Weight loss (%) = ( W_0_ − W_t_ ) × 100/Wt, where W_0_ is the intial weight at harvest and W_t_ is the weight of samples at different storage days.

### 4.5. Pigments Assay

The pigment contents of Chinese kale were determined by the previous method with some modifications [52]. Fresh Chinese kale tissue samples (0.5 g) were soaked with 25 mL of an acetone and ethanol solution (1:1, v:v) until the color faded to white (for 24 h). Then, the supernatant solution was detected at 663 nm, 645 nm, and 440 nm by a UV–Vis (Shimadzu UV-16A, Shimadzu, Corporation, Kyoto, Japan). Pigments were quantified as follows:Chlorophyll a (g kg^−1^) = (12.7 × OD_663_ − 2.69 × OD_645_) × V/W × 1000
Chlorophyll b (g kg^−1^) = (22.9 × OD_645_ − 4.86 × OD_663_) × V/W × 1000
Total Chlorophyll (g kg^−1^) = (8.02 × OD_663_ + 20.20 × OD_645_) × V/W × 1000
Carotenoids (g kg^−1^) = (4.7 × OD_440_ − 0.27 × Total Chlorophyll) × V/W × 1000
where V is the volume of the extract and W is the weight of sample.

### 4.6. Soluble Protein Content Assay

The soluble protein content of Chinese kale was determined by Coomassie blue staining [53]. Fresh samples (0.5 g) were ground with 8 mL of distilled water and then centrifuged at 4000× *g* for 15 min at 4 °C. The supernatant (0.5 mL) was diluted in the same volume of distilled water and well-mixed with 5 mL of Coomassie brilliant blue G-250 solution. After five minutes, the solution was measured at 595 nm by a UV–Vis.

### 4.7. Free Amino Acid Content Assay

The free amino acid content was measured by a Ninhydrin reaction [54]. Fresh samples (1.0 g) were ground with 5 mL of acetic acid, diluted to 0.1 L by distilled water, and then filtered with a long-necked funnel with double filter papers. 10^−3^ L of extracting solution, 2 mL of deionized water, 3-hydrated indene ketone solution, and 0.1 mL of ascorbic acid solutions were mixed together in order and then boiled for 15 m. Then, 60% ethanol was made up to 20 mL. The free amino acid content was measured at 570 nm by a UV–Vis.

### 4.8. Vitamin C Content Assay

Vitamin C content was determined spectrophotometrically using the Molybdenum Blue method with some modification [55]. Fresh samples (0.5 g) were grounded with 25 mL of 2% oxalic acid solution and then centrifuged at 7000× *g* for 10 min. Finally, supernatant (4 × 10^−3^ L) was mixed with 2 mL of sulfuric acid and 4 mL of ammonium molybdate and then measured at 500 nm by a UV–Vis.

### 4.9. Total Phenolic Content, Total Flavonoids Content and Antioxidant Activity Assay

Fresh powder samples (0.5 g) were soaked with 8 mL methanol for five minutes, then centrifuged 3000× *g* at 4 °C for 15 m. The supernatant was used for the following analyses.

The total phenolic content was measured according to Rahman [56]. Supernatant (0.5 mL) was treated with 0.5 mL of Folin–Ciocalteu’s phenol, and then 1.5 mL of 26.7% Na_2_CO_3_ and 7 mL of distilled water was added. The mixture was incubated at 50 °C for 5 m and immediately cooled to 25 °C. The absorbance was measured by a UV–Vis at 760 nm.

The total flavonoids content was measured by using the Al(NO_3_)_3_ colorimetric assay [57]. Sample extract (5 mL) were added into 0.5 mL of 30% methanol and 0.35 mL of 5% NaNO_2_ solution. The mixtures were blended and kept for 5 m at 25 °C. After that, 10% AlCl_3_ (0.35 mL) was added, and after the solution was kept still for 6 m 5 mL of 5% NaOH was added. The sample absorbance at 510 nm was measured by a UV–Vis.

The FRAP reducing antioxidant power assay was carried out according to Benzie and Strain [58]. After overtaxing, TPTZ solution (3.6 mL) was added into portions of 0.4 mL of the supernatant samples extract, which were then incubated at 37 °C for 10 m. The absorbances of the resultant solution were determined by a UV–Vis at 593 nm.

DPPH radical scavenging activity was determined according to Brand–Williams [59]. The sample extract (0.05 mL) reacted with 2.95 mL of DPPH solution for 15 m in the dark. The absorbance was measured at 515 nm by a UV–Vis.

### 4.10. Statistical Analysis

The measurements were conducted with three replications per treatment. Statistical analysis of data was performed with SPSS 23.0 software (SPSS Inc., Chicago, IL, USA). Significance among the treatments were determined by analysis of variance (ANOVA) followed by Duncan’s test. The figures were made by OriginPro 9.0 software (OriginLab Inc., Northampton, UK). TBtools software [60] was used for visualizing the transformed data into a cluster heatmap.

## 5. Conclusions

During storage, light exposure might retard plant senescence and lead to higher cost of production. However, pre-harvest supplemental blue light treatment of Chinese kale contributed to a shorter shelf-life; to higher contents of vitamin C, total chlorophyll, total soluble protein, and free amino acids; and to higher nutrition and antioxidant activity in the flower stalk of Chinese kale during storage. Since supplemental blue light could extend the freshness of Chinese kale, it could also reduce the transportation loss rate. Thus, pre-harvest supplemental blue light treatment in greenhouses might be a valuable method to enhance antioxidant activity, improve quality, and extend the shelf-life of the flower stalk in Chinese kale during storage.

## Figures and Tables

**Figure 1 plants-10-01177-f001:**
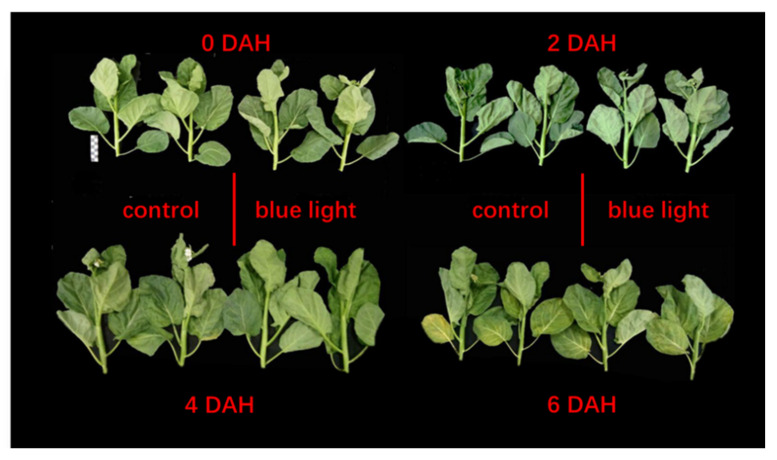
Flower stalk of Chinese kale samples in darkness at 15 °C.

**Figure 2 plants-10-01177-f002:**
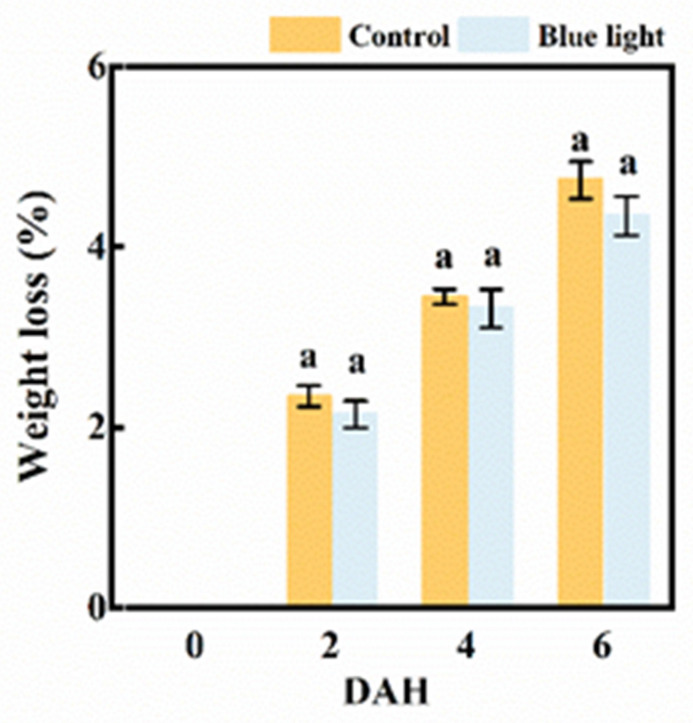
The weight loss of the flower stalk of Chinese kale during storage in darkness at 15 °C. Values with different lowercase letters on the top of the columns indicate significant differences (*p* < 0.05), according to Duncan’s test. Vertical bars represent the standard margin of error. 0 = 0 DAH, 2 = 2 DAH, 4 = 4 DAH, and 6 = 6 DAH.

**Figure 3 plants-10-01177-f003:**
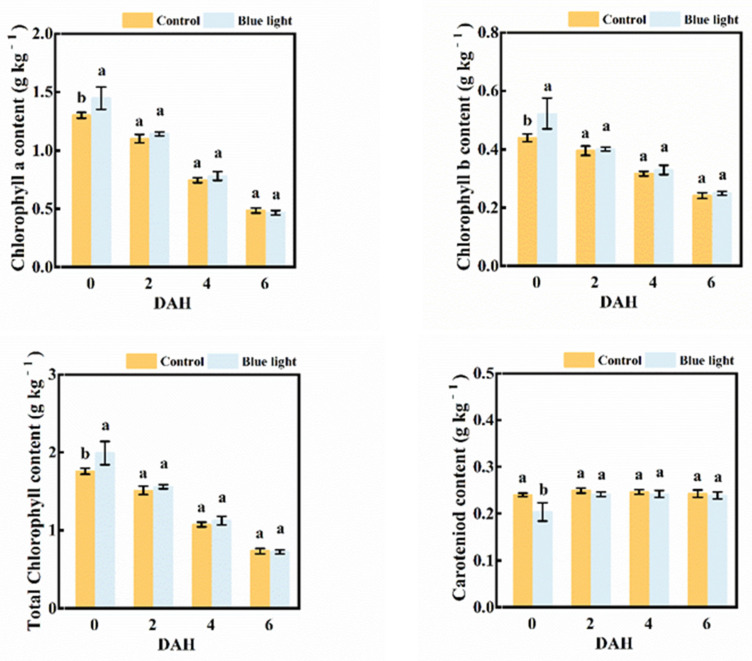
Pigment contents in the flower stalk of Chinese kale during storage in darkness at 15 °C. Values with different lowercase letters on the top of the columns indicate significant differences (*p* < 0.05), according to Duncan’s test. Vertical bars represent the standard margin of error. 0 = 0 DAH, 2 = 2 DAH, 4 = 4 DAH, and 6 = 6 DAH.

**Figure 4 plants-10-01177-f004:**
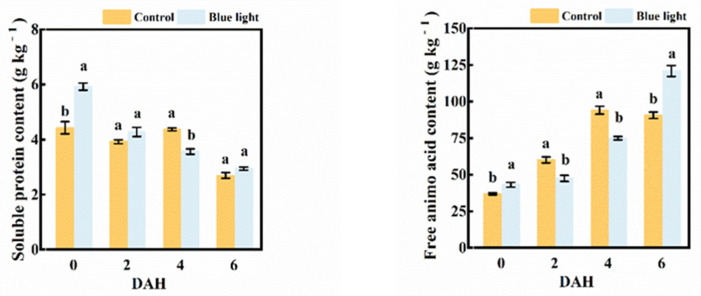
Soluble protein content and free amino acid content in the flower stalk of Chinese kale during storage in darkness at 15 °C. Values with different lowercase letters on the top of the columns indicate significant differences (*p* < 0.05), according to Duncan’s test. Vertical bars represent the standard margin of error. 0 = 0 DAH, 2 = 2 DAH, 4 = 4 DAH, and 6 = 6 DAH.

**Figure 5 plants-10-01177-f005:**
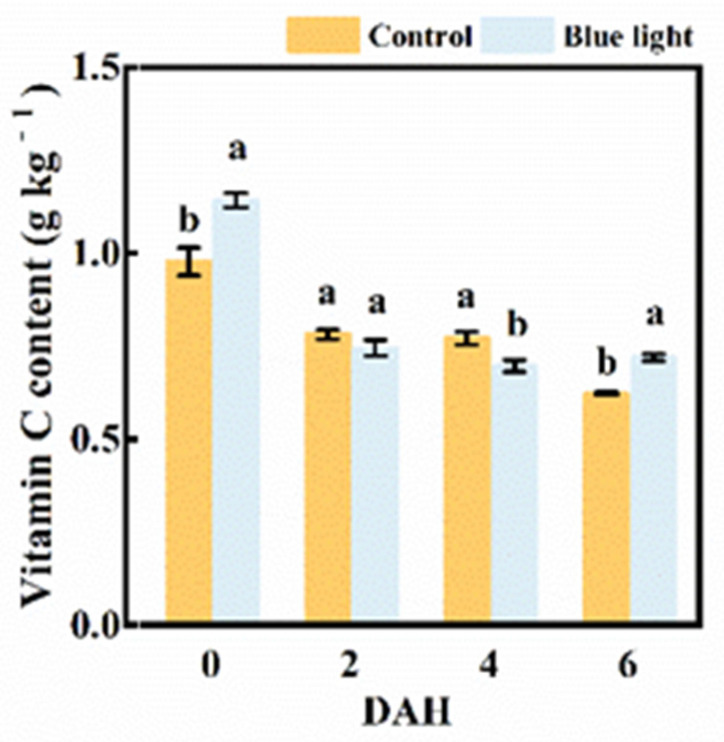
Vitamin C content in thew flower stalk of Chinese kale during storage in darkness at 15 °C. Values with different lowercase letters on the top of the columns indicate significant differences (*p* < 0.05), according to Duncan’s test. Vertical bars represent the standard margin of error. 0 = 0 DAH, 2 = 2 DAH, 4 = 4 DAH, and 6 = 6 DAH.

**Figure 6 plants-10-01177-f006:**
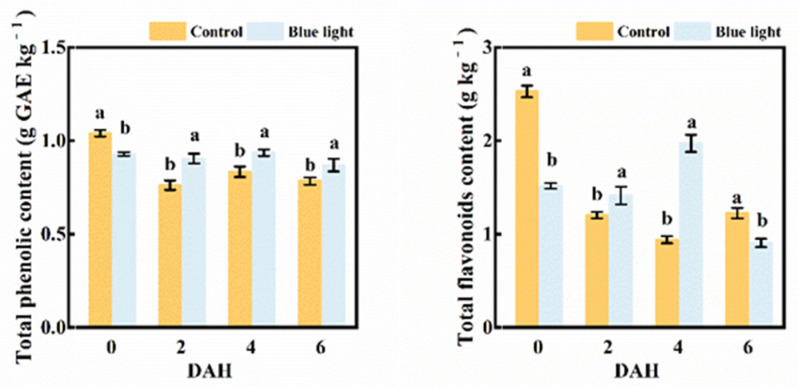
The contents of total phenolic and total flavonoids in the flower stalk of Chinese kale during storage in darkness at 15 °C. Values with different lowercase letters on the top of the columns indicate significant differences (*p* < 0.05), according to Duncan’s test. Vertical bars represent the standard margin of error. 0 = 0 DAH, 2 = 2 DAH, 4 = 4 DAH, and 6 = 6 DAH.

**Figure 7 plants-10-01177-f007:**
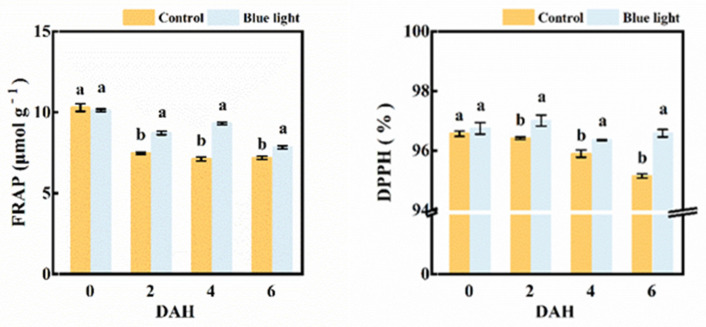
The FRAP value and DPPH activity in the flower stalk of Chinese kale during storage in darkness at 15 °C. Values with different lowercase letters on the top of the columns indicate significant differences (*p* < 0.05), according to Duncan’s test. Vertical bars represent the standard margin of error. 0 = 0 DAH, 2 = 2 DAH, 4 = 4 DAH, and 6 = 6 DAH.

**Figure 8 plants-10-01177-f008:**
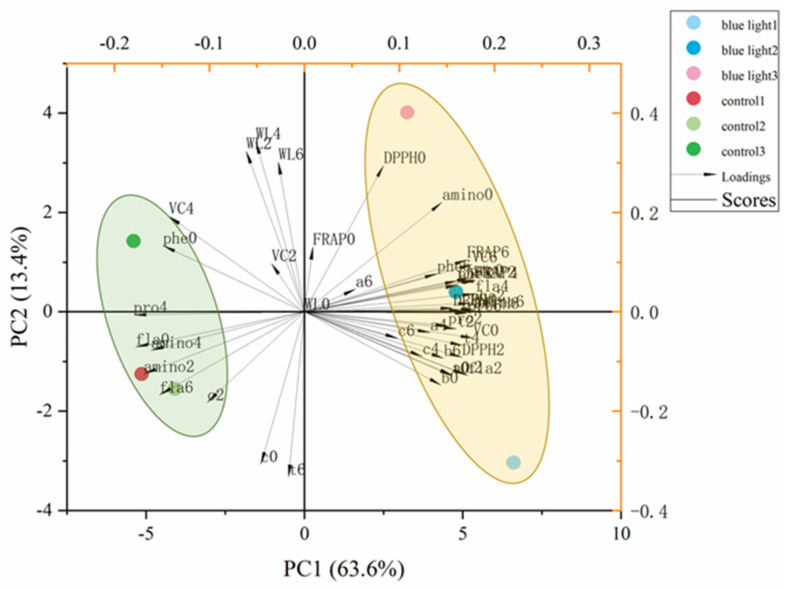
Multivariate principal component analysis showing pre-harvest supplemental blue light enhanced quality of the flower stalk in Chinese kale during post-harvest storage: a, chlorophyll a; b, chlorophyll b; t, total chlorophyll; c, carotenoids; WL, weight loss; phe, total phenolic content; fla, flavonoids; VC, Vitamin C; pro, soluble protein; amino, free amino acids; DPPH, DPPH radical inhibition percentage; FRAP, ferric ion reducing antioxidant power. 0 = 0 DAH, 2 = 2 DAH, 4 = 4 DAH, and 6 = 6 DAH.

**Figure 9 plants-10-01177-f009:**
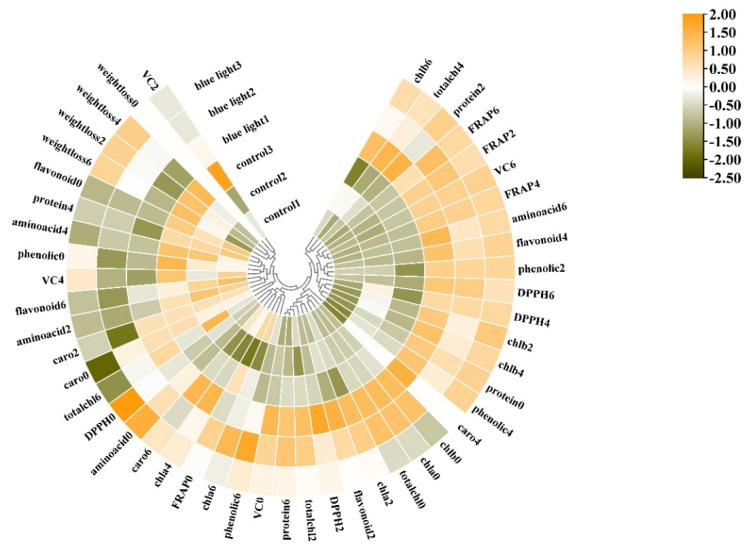
Cluster heatmap analysis summarizing the quality in flower stalk of Chinese kale in supplemental blue light and control treatment. Results are visualized using a false color scale with yellow as an increased parameter while green as a decreased parameter. chla, chlorophyll a; chlb, chlorophyll b; totalchl, total chlorophyll; caro, carotenoids; phenolic, total phenolic; flavonoid, total flavonoids; VC, Vitamin C; protein, soluble protein; aminoacid, free amino acids; DPPH, DPPH radical inhibition percentage; FRAP, ferric ion reducing antioxidant power. 0 = 0 DAH, 2 = 2 DAH, 4 = 4 DAH, and 6 = 6 DAH. The color scale from green to orange indicates the quality in the flower stalk of Chinese kale from low to high.

**Table 1 plants-10-01177-t001:** Eigen values, factor scores, and contribution of the five principal component axes to variation in the flower stalk of Chinese kale responses to supplemental blue light and control.

Principal Components	PC1	PC2	PC3	PC4	PC5
Eigen value	28.9	6.3	4.8	3.2	2.8
Variance (%)	62.9	13.7	10.4	7.0	6.0
Cumulation (%)	62.9	76.6	87.0	94.0	100.0

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
