# Peer review of "Pre-Harvest Supplemental Blue Light Enhanced Antioxidant Activity of Flower Stalk in Chinese Kale during Storage"

_plants, 2021, doi:10.3390/plants10061177_

Round 1

Reviewer 1 Report

Jiang et al. studied the effect of supplementary blue light during the cultivation period on the postharvest quality of Chinese Kale leaves. The topic is appropriate for Plants. The manuscript needs a substantial revision before it can be considered for publication in Plants. Specific points for revision follow bellow. Points for improvement Format according to the Plants instructions for authors.

Title: The results have shown a minimal effect of blue light, I propose the revision of the title as the effect was confined in the levels of antioxidants.

Lines 24-25. Please remove the reference about the specific illnesses, the argument stands fine as it is in lines 22-23.

Line 27 remove plants Line 27 revise “nutritional and commercial loss, so does the post-harvest Chinese kale”

Line 30 revise to “usually happen simultaneously” Line 39 be specific for vitamin fluctuation

Line 42 revise “single blue light”

Line 46 revise “post-harvest” broccoli to “broccoli during its post-harvest life” or something similar, do so throughout the manuscript. Post-harvest is not an adjective.

Line 51 revise “exhibited highly increases” to highly increased

Line 52 define accelerate phenolic accumulation

Line 58 revise plants to “plant products”

Line 72 revise nutrition to nutritional Move fig. 8 and table 1 to the supplementary data part of the manuscript. Figure 9 explain the treatments in the figure legend Discussion lines 216-231 the authors list results of other experiments and state that their treatments caused no significant impact. This part must be reduced in size.

Line 236-237 this part has no relevance to the findings of this study Again concerning the chl content the authors should cut down the discussion and focus in the significance of their results and the explanation of the phenomenon (initially higher Clh and lower Car following with no significant differences) as no effect on the quality actually exists.

Line 261 revise “kept higher” Line 265 this argument is not supported by the findings of this research Line 267 revise, the taste is influenced by the amino acid content, is not defended by this parameter.

Line 278 revise as the statement is wrong Line 280 revise “balance the level” Critical Point.

Line 309 provide a plausible explanation for these findings (for example is a priming effect possible that lead to the increased synthesis of antioxidants during the postharvest life?).

Line 318 revise previous result Line 335 provide the supplier of the lamps Critical point. 4.3 Why this temperature is selected. Provide reasoning. For postharvest storage the product is commonly stored at 1-4oC 378 revise Molybdenum Blue Spectrophotometry to determined spectrophotometrically using the Molybdenum Blue method Line 382 revise UV-spectrophotometer to UV-Vis (as 500 nm belongs in visual part of the spectrum) Major point.

In general the description of the methods needs improvement.

Reviewer 2 Report

The matter is well organized and the paper well written. Some minor suggestions should be addressed:

The abstract should be written in a clearer manner

A graphical scheme should be inserted for describing: Plant material and cultivation conditions, Supplemental blue light treatment and Storage 

Additional lines for describing the updated approach of study of antioxidant properties should be added and related references such as:

Durazzo A. Study Approach of Antioxidant Properties in Foods: Update and Considerations. 02/2017 Foods; 6(3):17., DOI:10.3390/foods6030017

Apak R., Gorinstein S., Böhm V., Schaich K.M., Özyürek M., Kubilay Güçlü K. Methods of measurement and evaluation of natural antioxidant capacity/activity (IUPAC Technical Report) Pure Appl. Chem. 2013;85:957–998. doi: 10.1351/PAC-REP-12-07-15.

Figure 8 and 9 should be better described in the text

Limits, advantages, practical applications and future directions should be marked in Conclusion

Round 2

Reviewer 1 Report

Jiang et al presented an updated version of their manuscript that is significantly improved. However, a couple of points remain that they should be addressed before the publication of the article. 

1) The article is not yet formatted according to the Plants instructions for authors (both the references inside the manuscript and the reference list).

2) in my point for improvement 14. (Line 309) I asked the authors to provide a plausible explanation for their findings, the explanation was not included in the manuscript.  

3) Point 18. I didn't challenge the method that the authors used, I asked them to revise the description of the method in the corresponding part. Please check again. 
